# Thrombocytopenia in Virus Infections

**DOI:** 10.3390/jcm10040877

**Published:** 2021-02-20

**Authors:** Matthijs Raadsen, Justin Du Toit, Thomas Langerak, Bas van Bussel, Eric van Gorp, Marco Goeijenbier

**Affiliations:** 1Department of Viroscience, Erasmus MC Rotterdam, Doctor molewaterplein 40, 3015 GD Rotterdam, The Netherlands; m.p.raadsen@erasmusmc.nl (M.R.); thomas.langerak@erasmusmc.nl (T.L.); e.vangorp@erasmusmc.nl (E.v.G.); 2Department of Haematology, Wits University Donald Gordon Medical Centre Johannesburg, Johannesburg 2041, South Africa; justinr_dutoit@yahoo.com; 3Department of Intensive Care Medicine, Maastricht University Medical Center Plus, 6229 HX Maastricht, The Netherlands; bas.van.bussel@mumc.nl; 4Care and Public Health Research Institute (CAPHRI), Maastricht University, 6229 GT Maastricht, The Netherlands; 5Department of Internal Medicine, Erasmus MC Rotterdam, 3000 CA Rotterdam, The Netherlands

**Keywords:** virus infection, thrombocytopenia, thrombocytopathy, aggregation, HIV, SARS-CoV-2, hantavirus, coronavirus, influenza

## Abstract

Thrombocytopenia, which signifies a low platelet count usually below 150 × 10^9^/L, is a common finding following or during many viral infections. In clinical medicine, mild thrombocytopenia, combined with lymphopenia in a patient with signs and symptoms of an infectious disease, raises the suspicion of a viral infection. This phenomenon is classically attributed to platelet consumption due to inflammation-induced coagulation, sequestration from the circulation by phagocytosis and hypersplenism, and impaired platelet production due to defective megakaryopoiesis or cytokine-induced myelosuppression. All these mechanisms, while plausible and supported by substantial evidence, regard platelets as passive bystanders during viral infection. However, platelets are increasingly recognized as active players in the (antiviral) immune response and have been shown to interact with cells of the innate and adaptive immune system as well as directly with viruses. These findings can be of interest both for understanding the pathogenesis of viral infectious diseases and predicting outcome. In this review, we will summarize and discuss the literature currently available on various mechanisms within the relationship between thrombocytopenia and virus infections.

## 1. Introduction

In patients presenting to care with signs or symptoms of infectious disease, a full blood count is part of a routine diagnostic evaluation. Mild thrombocytopenia, often combined with lymphocytopenia is typical of most acute viral infections, but neither are sufficiently sensitive nor specific to reliably distinguish viral from bacterial or parasitic pathogens. Except for viral hemorrhagic fevers and rare cases of severe disseminated viral infections, virus-induced thrombocytopenia does not lead to significant bleeding, rarely requires platelet transfusions, and is therefore easily dismissed as clinically irrelevant. However, when the relationship between platelets and viral infection is studied more closely and in larger study populations, important findings emerge which shed light on previously unrecognized aspects of viral diseases. The incidence of thromboembolic complications is elevated in individuals during and after influenza virus infection, for example, a relation which may not be apparent to physicians diagnosing and treating influenza-like illness [1]. Platelet counts during peak symptomatic disease have also been found to be a marker of disease severity in certain viral infections, [2,3,4,5,6,7,8,9,10,11,12,13,14,15,16,17,18,19] or can serve as a first clue towards diagnosing chronic viral infections [20,21,22,23]. These phenomena are typically not explained by changes in platelet quantity, but rather by the effects of viral infections on platelet function.

Platelets are small, anucleate cells that circulate in the blood for approximately 7 to 10 days after being formed. Their main physiological role is hemostasis, forming blood clots (thrombi) to safeguard vascular integrity. Platelets originate from megakaryocytes, which are giant polyploid cells residing in the bone marrow that have themselves formed from hematopoietic stem cells. Megakaryocytes develop proplatelets that bud off numerous platelets into the blood stream, after endoplasmic maturation [24]. In individuals with normal bone marrow function, platelets circulate at levels between 150 to 450 × 10^9^/L [25]. Megakaryopoiesis is stimulated by a number of cytokines, with Stromal Derived Factor 1 (SDF-1), Granulocyte-Monocyte Colony Stimulating Factor (GM-CSF) Interleukins (IL-3, IL-6, and IL-11) Fibroblast Growth factor 4 (FGF-4) and thrombopoietin (TPO) being the most important [26]. Whereas TPO plays a crucial role in maintenance of hematopoietic stem cells [27], most of these cytokines are proinflammatory and induce rapid maturation and activation of leukocytes, as well as stimulating megakaryopoiesis, which illustrates how platelet production is affected by inflammatory processes.

Conversely, platelets also affect the inflammatory response to viral infection and can even internalize several viruses directly. In response to infection, platelets interact with leukocytes and vascular endothelial cells before activating and secreting soluble prothrombotic and inflammatory mediators stored within granules [28]. Despite not having a nucleus, platelets do contain some RNA and maintain a limited ability for protein translation, enabling some regulation of this response [29], but also potentially supporting replication of some RNA viruses [30].

In this review, all of the literature on the relationship between platelets and viral infectious diseases published between 2010 and late 2020 has been systematically assessed and summarized in a narrative format, classified per virus category.

## 2. Search Strategy

The online database from the national Center of biotechnology information, Pubmed, was queried on August 3rd 2020 using the search term available in the supplemental information section included with this paper. This search was repeated on December 21st 2020, in order to include the latest publications on the currently pandemic Severe Acute Respiratory Syndrome Coronavirus 2 (SARS-CoV-2).

Additional filters were applied to restrict results to original research papers published during the previous 10 years, related to humans and with full text available. Individual papers were assessed by the first author, based on title and abstract for relevance, originality and quality and sorted based on virus species or virus family. The findings of the papers that are relevant for the subject matter of this review were presented in a narrative format within the following subgroups: “General Topics”, “Arboviruses”, “Blood Borne Viruses”, “Rodent Borne Viruses”, “Gastrointestinal Viruses”, “Herpesviruses” and “Respiratory tract Viruses”. Additional references were added to provide more context when necessary.

## 3. Results

The search yielded 413 papers. The results of the author classification process are summarized in Figure 1.

### 3.1. Platelets and Viral Infections: General Principles

An overview of the mechanisms that contribute to thrombocytopenia in the main viral infections discussed in this review, including references, can be found in Table 1. 

#### 3.1.1. Aggregation

Platelet agglutination or adhesion to leukocytes is often found in patients with systemic inflammatory diseases, including viral infections. Standard automated hematology analyzers are often unable to accurately detect leukocyte-bound platelets or platelet aggregates, leading to a false finding of a reduced platelet count (pseudo-thrombocytopenia). This can also be caused by drawing blood into tubes containing EDTA, the most common anticoagulant used for complete blood counts. If pseudo-thrombocytopenia is suspected, performing a manual peripheral blood smear or repeating the platelet count in using different anticoagulants, can help avoid unnecessary diagnostic procedures or transfusions [31]. Isolated thrombocytopenia should prompt investigation into chronic viral infections, such as hepatitis B, C and HIV, whereas leukocyte abnormalities and rise in other infection biomarkers, raises suspicions of an acute viral illness [32].

#### 3.1.2. Impaired Hematopoiesis

One of the most common accepted etiologies underlying virus-induced thrombocytopenia is the one where viruses can directly infect bone marrow stromal cells and hematopoietic stem cells leading to defective hematopoiesis and thrombocytopenia [33]. Furthermore, decreased platelet production can be the result of changed cytokine profiles, during infection, leading to lower TPO production in the liver, reducing megakaryopoiesis [34,35]. Finally, viruses can infect and replicate in megakaryocytes while other viruses modulate megakaryocyte function or decrease the expression of Myeloproliferative Leukemia Protein (c-MPL), the receptor for TPO, leading megakaryocyte destruction and subsequent lowered platelet production [36,37,38].

#### 3.1.3. Sequestration and Intravascular Destruction

Platelet destruction can occur via direct interaction of platelets with viruses. This interaction occurs via a range of receptors including Toll-Like Receptors (TLRs), integrins (GPIIb/IIIa) and c-type lectins (CLEC), that interact with the different viruses leading to platelet activation, degranulation and clearance in liver and spleen [39,40,41]. Upon viral infection the host defense generally induces a systemic inflammatory response, which leads to platelet activation and subsequent clearance [42]. Furthermore, platelets can bind to neutrophils, forming platelet-neutrophil aggregates, which in turn triggers the phagocytosis of platelets [43,44]. Additionally, many viruses can activate the coagulation system by induction of tissue factor leading to thrombin generation and platelet activation with subsequent platelet clearance via protease activating receptor (PAR) signaling. PARs are present on platelets, leucocytes and endothelial cells which modulate the innate immune responses [45]. Platelets can bind immunoglobulins (attached to viruses) via Fc-gamma-RII receptors leading to platelet activation, aggregation and clearance [46,47], while immunoglobulins produced by B-lymphocytes that target viruses can cross-react with platelet surface integrins (GPIIb/IIIa or GPIb-IX-V) leading to immune thrombocytopenia (ITP) [48]. In depth, more virus specific mechanisms will be discussed in the specific virus groups.

#### 3.1.4. Platelet Expression of Pattern Recognition Receptors (PRR)

PRRs such as TLRs and CLECs, [49] or messenger ribonucleic acids (mRNAs) can identify pathogen associated molecular patterns (PAMPs) from viruses and many are expressed by platelets [44]. This direct interaction of a virus or its genome with PRR can lead to platelet activation and subsequent release of chemokines. This enhances endothelial cell signaling, leucocyte migration and direct interaction and activation of leucocytes [50]. These complex interactions may have both an immune protective mechanism, [44] or be injurious to the host.

#### 3.1.5. Platelets Can Induce Inflammation and Secrete Anti-Microbial Proteins

Activated platelets undergo degranulation and release numerous inflammatory mediators, cytokines and chemokines stored in granules. Three types of granules exist: α granules, dense granules and lysosomal granules. These granules contain different molecules that can exert pro-thrombotic and immune effects leading to direct and indirect interactions with different pro-inflammatory immune cells, causing a local or systemic inflammatory milieu [51]. In addition, the α-granules can secrete platelet microbicidal peptides (PMPs) that have direct anti-viral effects, for example, it has previously been shown that synthetic PMPs have strong viricidal effects against vaccinia virus [52].

#### 3.1.6. Platelets Act as Antigen Presenting Cells (APCs)

APCs require MHC-I molecules to present antigen to CD8+ T cells. There is evidence now that platelets and megakaryocytes contain all the MHC-I and co-stimulatory molecules necessary for antigen presentation including the entire proteome [53,54]. In addition, it has been shown that platelets can successfully present ovalbumin and malarial antigens to activate CD8+ T cells [55].

### 3.2. Arboviruses

Arthropod borne viruses (arboviruses) are viruses that are transmitted to humans through arthropods, mainly mosquitoes and ticks. Some of these arboviruses, especially flaviviruses, potentially cause hemorrhagic fever. In this section, we will discuss the most prevalent arboviruses as well as potentially emerging arboviruses.

#### 3.2.1. Dengue Virus

With an estimated 390 million annual infections, dengue virus (DENV) is the most prevalent arbovirus worldwide [56]. Thrombocytopenia is a hallmark of severe DENV infection and platelet levels are lower in DENV infected patients compared to other febrile illnesses [57]. Severe thrombocytopenia <20 × 10^9^/L occurs frequently in hospitalized dengue patients, and is associated with prolonged admission, plasma leakage and the presence of clinical warning signs [58,59]. Severe DENV infections with severe thrombocytopenia, hemorrhage and plasma leakage, occur more often in secondary DENV infection compared to primary DENV infection, due to antibody-dependent enhancement [60,61,62]. Below, the many mechanisms for thrombocytopenia in DENV disease that have been identified will be discussed.

Platelets can directly be activated by circulating DENV particles and by immune factors released during the acute phase of DENV infection. This activation induces upregulation of platelet adhesion molecules on the surface of vascular endothelial cells, causing more indirect platelet activation [63,64]. Activated platelets undergo degranulation, attach themselves to the vascular wall and form thrombi, effectively removing them from the circulation, which results in thrombocytopenia. Despite this activation, platelets from dengue patients are hyporesponsive to procoagulant stimuli in aggregometry assays, which is likely the result of exhaustion [65]. This illustrates thrombocytopenia and platelet dysfunction go hand in hand during DENV infection. The DENV NS1 protein interacts with both TLR 4 and TLR 2, expressed by platelets, leading to platelet activation, aggregation, adherence to endothelial cells and phagocytosis by macrophages [66]. DENV can also activate platelets via CLEC2, that in turn stimulate macrophages and neutrophils via CLEC5A and TLR 2. Activated neutrophils subsequently form Neutrophil Extracellular Traps (NETs), providing a scaffold for prothrombotic factors, such as platelets, red blood cells and molecules involved in both the intrinsic and extrinsic coagulation pathways. NETs also activate platelets via TLR-4, creating a positive feedback loop between NETosis and platelet activation. These mechanisms contribute to thrombocytopenia, bleeding, vascular leakage and lethality in mouse models of dengue hemorrhagic shock syndrome [66,67]. The formation of platelet-monocyte aggregates in DENV infection has also been demonstrated and correlates with thrombocytopenia and clinical signs of vascular leakage [68,69]. These platelets alter the monocyte’s innate immune response, by inducing production of IL-10 [68] and inhibiting the production of interferon α [70]. Serotonin enhances DENV-mediated platelet activation and is released by perivascular mast cells in a mouse model of DENV infection, leading to thrombocytopenia, suggesting serotonin-blocking drugs might be beneficial [71]. Platelet activating factor (PAF) is another inducer of platelet aggregation and vascular leakage elevated in plasma of acute dengue patients. A recent small phase II randomized placebo-controlled clinical trial showed that Rupatadine, a licensed antihistamine with PAF receptor blocking activity, is safe in acute dengue patients, but not clinically beneficial, with no effect on nadir platelet counts [72].

In response to DENV infection, the immune system also depletes platelets through direct cytotoxic effects. This occurs through complement factor C3 binding to the platelet surface and opsonization of platelets by DENV NS1 specific IgG, leading to subsequent phagocytosis by macrophages [73,74]. The observation that patients with severe DENV infection have increased afucosylated IgG1 antibodies and that these antibodies can cross-react with platelet antigens, further supports the notion that platelet-NS1 cross-reactive antibodies contribute to the depletion of platelets during severe DENV infection [75]. Infusion of anti-NS1 IgG cannot elicit thrombocytopenia in mouse models. However, it does enhance thrombocytopenia when administered during DENV infection. This could indicate that platelet binding and destruction is mediated by IgG bound NS1 dimers, rather than a direct interaction between cross-reactive IgG and platelets. This could have implications for vaccine design [76,77]. Platelets upregulate their expression of Human Leukocyte Antigen (HLA) class I molecules in the presence of DENV, suggesting a possible role in viral antigen presentation and T cell mediated cytotoxicity as a mechanism for thrombocytopenia [64].

A third mechanism is direct infection of platelets and megakaryocytes by DENV, leading to viral replication, cell lysis and impaired production of platelets. DENV infects platelets by binding DC-SIGN (CD209) and heparan sulfate proteoglycan (HSP) on the platelet surface and indirectly via Fc receptor FcϒR2A (CD32), after binding of antibodies to the virus particle [30]. The presence of DENV specific antibodies is not required for platelet infection [78]. Especially during the viremic stage and in those patients with more severe dengue, less DC-SIGN and FcϒR2A expression is detected on platelets when compared to patients suffering from other febrile illnesses. The relation between this phenomenon with thrombocytopenia has not been explored however and it is unclear whether the presence of DENV particles in the blood could interfere with the binding of the detection antibodies. There could also be a survival advantage of platelets with low DC-SIGN and FCϒR2A expression [79]. DENV productively infects a megakaryoblast-like cell line, using Glycoprotein Ib (GPIb) and was found circulating megakaryocyte-like cells in Rhesus Macaques [80]. This suggests that DENV can infect mature megakaryocytes and impair platelet production and survival through replication, although this has not been confirmed in vivo [81].

#### 3.2.2. Chikungunya virus (CHIKV)

Despite sharing many clinical characteristics with DENV, thrombocytopenia has only been occasionally described in CHIKV infected patients [82]. In a cross-sectional study of arbovirus infections in Pakistan, thrombocytopenia was observed in 18% of CHIKV infections, compared to 74% in DENV. Furthermore, thrombocytosis was significantly associated with CHIKV infection (OR 2.2) [83].

#### 3.2.3. Crimean Congo Hemorrhagic Fever (CCHF) 

CCHF is a tick-borne virus from the Bunyaviridae family that is associated with pronounced thrombocytopenia, accompanied by bleeding complications [84]. CCHF cases with bleeding have lower platelet counts and slightly raised platelet distribution width compared to those who do not bleed [85]. A lower platelet lymphocyte ratio (PLR) upon presentation to hospital care is predictive of adverse clinical outcomes in CCHF patients [6].

#### 3.2.4. Japanese Encephalitis Virus (JEV) 

This vaccine preventable flavivirus has a wide distribution in eastern Asia. Although central nervous system infection has a substantial mortality rate of 20–30%, most cases are either asymptomatic or mild, without clinically overt signs of encephalitis [86]. A recent prospective study performed in hospitalized patients with dengue-negative febrile illness in Indonesia found that 6% had serological evidence of recent JEV infection. Thrombocytopenia was common during the acute phase of illness in these non-encephalitic cases (69%) but did not occur as frequently as dengue cases from the same study (92%) [87].

#### 3.2.5. Severe Fever with Thrombocytopenia Syndrome (SFTS) 

SFTS is an emerging infectious disease caused by a recently discovered Bunyavirus that, as the name suggests, is associated with profound thrombocytopenia [88]. DC-SIGN, which is expressed by human platelets, was identified as a receptor for Bunya viruses, but no studies have shown virus entry or replication of SFTS in platelets [89]. A recent detailed study in Chinese SFTS patients revealed a severity-dependent depletion of the essential amino acid arginine due to arginase released by granulocytic Myeloid-derived suppressor cells (gMDSC’s), which are recruited during the acute stage of the disease. The resulting lack of both arginine and its metabolite, nitric oxide, are believed to disinhibit platelet activation, leading to platelet aggregation, destruction and thrombocytopenia. A randomized, non-controlled clinical trial was performed were arginine supplementation with best supportive care was compared with supportive care alone. Whilst not demonstrating a statistically significant survival benefit, platelet counts returned to normal more rapidly in the treatment arm [90].

#### 3.2.6. Tick-Borne Encephalitis Virus (TBEV) 

TBEV is a flavivirus that is transmitted through bites of ticks and is prevalent in Europe and northeastern Asia [91]. After initial viremia, the central nervous system may become infected, which can result in severe neurological damage and occasionally death. Like most other viruses that can enter the bloodstream, TBEV causes thrombocytopenia, albeit generally mild and not typically associated with bleeding. A case–control study in patients with suspected and confirmed central nervous system infections, reported a decreased mean platelet count in TBE cases versus neuroborreliosis cases. However, the mean platelet count (173.8 × 10^9^/L) remained above the lower limit of normal in the TBE group. In addition, platelet counts correlated positively with concentrations in serum of IL-23, a cytokine secreted by dendritic cells, which is believed to stimulate essential host-defense mechanisms against viruses [92]. This could indicate a role for IL-23 in stimulating megakaryopoiesis during TBEV infection.

#### 3.2.7. Viral Hemorrhagic Fevers (VHF)

The extreme containment precautions required to study the most highly pathogenic VHF’s, such as Ebolavirus, Marburgvirus and Lassavirus, makes studying the interactions of live virus with human platelets an expensive and labor intensive endeavor. Only a small number of laboratories worldwide are equipped with biosafety level 4 facilities and only two are located on the African continent, where most VHF infections occur [93]. Lymphocytic Choriomeningitis Virus (LCMV) is an arenavirus, which has sporadically caused severe Lassavirus-like illness in humans. Its reservoir host is mice, who only develop very mild disease. Experimental infections of mice with LCMV, after platelet depletion treatment, however, result in a more severe VHF with uncontrolled viral replication and dissemination, similar to that observed in humans. The platelet-depleted mice had impaired LCMV specific CD8+ T Cell responses. Severe disease and mortality only occurred in mice whose platelet reduction treatment was initiated shortly before LCMV infection. This may suggest that the innate antiviral response against LCMV requires platelets. However, Interferon α and β production appeared to be unaffected by platelet depletion treatment, indicating this part of the innate immune response had remained intact. Further examination of the spleen in LCMV infected mice who underwent platelet depletion demonstrated extensive disruption of the splenic architecture and cellular necrosis, which could be the common mechanism of both the defective innate and T cell responses. These experiments provided novel insights in the role of platelets in controlling VHF through protection of splenic vascular integrity and the importance of this organ in mounting sufficient cellular and innate immune responses to eliminate the virus [94].

#### 3.2.8. West Nile Virus (WNV) 

West Nile Virus (WNV) is a neurotropic flavivirus that mainly causes asymptomatic infections, while severe complications such as meningitis or encephalitis can occur, but are rare. WNV infections with hemorrhagic complications have sporadically been reported but severe thrombocytopenia and hemorrhage typically do not occur during WNV infection [95]. A case of WNV transmission through a platelet transfusion unit from a donor that tested negative on whole blood on the day of transfusion has been described, raising the possibility that the virus may concentrate in platelets, although the platelet unit itself was never tested [96].

#### 3.2.9. Yellow Fever Virus (YFV)

Yellow fever virus (YFV) is a flavivirus that can be transmitted through Aedes mosquitoes and other vectors and is endemic in Africa and the Americas but not in Asia [97]. Even though there is an effective vaccine against YFV, annually several tens of thousands of persons die of YFV infection in Africa [98]. Disseminated yellow fever is characterized by hepatitis and hepatic failure, with resulting thrombocytopenia, deficiencies in plasma coagulation factors, prolonged activated Partial Thromboplastin Time (aPTT) and an elevated International Normalized Ratio (INR) [99,100]. A Brazilian retrospective cohort study of patients suffering severe YFV infections, showed severe hemorrhagic complications, mostly from the gastrointestinal tract. Despite this, only mild thrombocytopenia was present, with a median platelet count of 74 × 10^9^/L, indicating platelet consumption as a result of hemorrhage, rather than thrombocytopenia-induced bleeding [100]. This is supported by the presence of ischemic and hemorrhagic microvascular pathology upon fundoscopic examination of hospitalized yellow fever patients, which correlate with the degree of thrombocytopenia present, disease severity and markers for renal and hepatic disease [101].

#### 3.2.10. Zika Virus (ZIKV) 

Like DENV, ZIKV is a flavivirus that is mainly transmitted through Aedes mosquitoes. In 2015/2016, there was a large outbreak of ZIKV in the Americas, which led to an increase in the incidence of Guillain-Barré syndrome in adults and congenital abnormalities in newborns [102]. Seroprevalence studies indicate that, depending on the country and location, up to 60% of inhabitants got infected with ZIKV during this outbreak [103,104,105]. A large prospective cohort study of ZIKV infected patients in Puerto Rico identified thrombocytopenia (defined in this study as a platelet count below 100 × 10^9^/L) in 1.2% of confirmed cases. Only 25% of those had platelet counts below 20 × 10^9^/L and 16% had etiologies other than ZIKV infection [106]. Platelet counts below 150 × 10^9^/L are present in a small minority of ZIKV infections, although profound thrombocytopenia has been described in fulminant cases, associated with bleeding, liver failure and other coagulation disorders [107,108]. Most ZIKV patients have platelet counts between 150 and 300 × 10^9^/L [109,110,111]. Thrombocytopenia on first diagnosis was significantly associated with higher odds of hospitalization in a case series study of U.S. veterans with laboratory-confirmed or presumptive ZIKV infection (OR 6.4, 95% CI 4.0–10.1) [3]. Suspected diagnosis of de novo Immune Thrombocytopenic Purpura (ITP) has been reported in patients presenting with severe thrombocytopenia during ZIKV convalescence, who responded to treatment with corticosteroids, intravenous immunoglobulin (IVIG), or both [112,113,114]. Exacerbation of pre-existing ITP during the acute phase of ZIKV disease has also been described [115]. ZIKV infects monocytes, macrophages and dendritic cells, but is unable to propagate in megakaryocyte-differentiated human hematopoietic stem cells and platelets [116]. Thrombocytopenia presents more often in DENV infections compared to ZIKV and can be used to distinguish the two otherwise similar diseases to a limited extent [117,118,119].

### 3.3. Blood-Borne Viruses

#### 3.3.1. Hepatitis B and C (HBV and HBC)

Chronic hepatitis B and C virus infections are associated with hepatic cirrhosis, portal hypertension, liver failure and hepatocellular carcinoma (HCC). Thrombocytopenia is a predictor of such adverse outcomes, especially when combined in clinical scoring systems with other parameters such as age, gamma-glutamyl transpeptidase, Alanine and Aspartate aminotransferase [7,8,9,10,13,14,15,17,19,120,121,122] However, in individuals that have already developed HCC, thrombocytosis carries a negative prognosis rather than thrombocytopenia. Elevated platelet counts, when combined in a ratio with lymphocyte counts (PLR), is a biomarker of malignant inflammation and elevation is predictive of a poor prognosis in patients with HCC [123,124]. However, this marker does not appear to have much added value when tumor size and histological parameters are known or in patients with either no cirrhosis of very severe cirrhosis [11,125].

A likely explanation for the lower platelet counts observed in patients with hepatic cirrhosis, including those with a non-viral etiology, is portal hypertension leading to increased sequestration of platelets and red blood cells in the liver and spleen [126].

Another factor more specific to HBV and HCV infection could be increased platelet activation and consumption due to chronic inflammatory processes caused by these viruses. Mean platelet volume (MPV), a biomarker of platelet activation, was found to be increased in patients with chronic hepatitis B infection, compared to controls with no documented HBV infection [127]. Chronic HBV infection is also associated with a low platelet response to clopidogrel, which could be due to platelet activation by the virus or changes in drug metabolism caused by liver disease [128].

HBV and hepatitis Delta (HDV) co-infection also appears to lower platelet counts compared to HBV mono-infection, independently of the severity of liver disease, though the mechanism is unknown [129].

In hepatitis C infection, intravascular destruction of platelets is likely also mediated by anti-platelet antibodies and ITP related to HCV has been reported in the literature [130]. In HCV infected patients, platelet auto-antibodies were more commonly identified as a cause in mild (platelet count 126–149 × 10^9^/L) thrombocytopenia, whereas decreased megakaryopoiesis was a more prominent contributing factor in more severely (platelet count < 100 × 10^9^/L) thrombocytopenic patients [131].

Since both hepatitis B and C are blood-borne pathogens, a direct interaction between these viruses and platelets is possible. Hepatitis C virus RNA appears more stable when incubated with platelets than with platelet-free plasma, although there is no evidence of replication [132]. Instead, platelets may shield the virus from complement, interferons, cytokines, antibodies or other antiviral factors present in serum, perhaps through internalization. There have been reported cases of HCV infected patients with much higher viral loads in the platelet compartment compared to plasma [130]. One study found that, while HCV viral load was higher in plasma compared to platelets in patients before interferon/Ribavirin treatment, some patients had detectable RNA in platelets after treatment, but not in serum, resulting in relapse [133]. This could indicate that platelets and their megakaryocyte precursors can provide a site for therapeutic drug and immune evasion, either perhaps due to diminished response to interferons or a relative impermeability for nucleoside analogues or both, while extrahepatic replication is of minor importance during untreated HCV infection. The therapeutic relevance of this finding today is questionable however, given the high curation rate that can be achieved using the novel direct antiviral agents that have become available. The effect of direct interactions between HCV and platelets on platelet lifespan remains to be established.

#### 3.3.2. Human Immunodeficiency Virus (HIV)

Thrombocytopenia is a common finding in HIV infection and HIV testing is part of a routine evaluation for unexplained thrombocytopenia. Screening individuals with thrombocytopenia and other so-called indicator conditions, has been shown to be a cost-effective and more efficient compared to universal screening [20,21,22]. HIV as the underlying cause for otherwise unexplained thrombocytopenia is frequently missed [23]. Decline of platelet count over time in HIV patients has been associated with development of dementia and reduced gray matter volume on MRI scans, although univariate and multivariate analyses were not entirely consistent [134]. Several mechanisms for thrombocytopenia in HIV infected individuals have been proposed and these will be discussed in more detail below.

Chronically infected HIV patients, even when adequately suppressed with combination Anti-Retroviral Therapy (c-ART), have a substantially increased risk of developing cardiovascular disease and deep venous thrombosis compared to HIV negative individuals. Platelet activation is believed to be a major driving factor behind this phenomenon. Untreated HIV positive individuals have elevated platelet activation markers in plasma compared to matched healthy controls, which positively correlate with viral load and negatively with CD4+ T cell counts. In addition, platelets from HIV patients express more oxidative stress-related proteins [135]. Studies investigating platelet morphology (MPV, PDW) in HIV patients, report contradictory results [136,137]. Platelets can bind HIV-1 virus through DC-SIGN and serve as a carrier, protecting virions in the circulation from antiviral factors through formation of RBC-platelet aggregates [138]. Platelets may also become activated after binding to HIV and release platelet factor 4 (PF4/CXCL4), from their α granules, which has been shown to inhibit cell attachment and entry of several HIV-1 strains through binding of its main envelope glycoprotein, GP-120 [139]. PF4 and other α-granule cytokines such as CCL5 likely also contribute to a chronic proinflammatory state in HIV patients. Interestingly, HIV patients on c-ART who become infected with DENV, were found to have milder disease and their platelets released less proinflammatory PF4 and CCL5 from α granules, likely resulting from previous HIV-associated depletion [140].

Very recently, platelets from long-term virologically suppressed HIV-infected individuals were shown to contain infectious HIV-1, demonstrating its ability to infect platelets and remain viable. Both viral RNA and proviral DNA of HIV was detected in bone marrow megakaryocytes from these same individuals, indicating platelets may already be infected upon formation. Platelets phagocytosis by macrophages was shown to lead to productive infection in this cell type [141]. The effects of HIV-1 infection on the activation status or lifespan of platelets were not investigated, meaning the role this mechanism plays in thrombocytopenia in people living with HIV remains unclear.

Another factor contributing to the prothrombotic state of HIV patients is the induction of platelet-monocyte aggregates (PMA) by the virus, which causes mutual activation of these two cell types [142,143]. Monocytes expressing CD16 (nonclassical and intermediate monocytes) appear to be primarily involved in the formation of PMA [144]. Electron microscopy (EM) studies of PMA in HIV infected individuals show a maximum of 4 simultaneously attached platelets per monocyte [145]. However, a single monocyte is likely capable of transiently interacting with many more platelets during its lifetime, leading to platelet activation and cumulative loss over time. The formation of PMA is mediated by binding of P-selectin on the platelet surface with P-Selectin Glycoprotein Ligand-1 (PSGL-1) on the monocyte surface. This leads to monocyte activation, including secretion of proinflammatory cytokines and Tissue Factor (TF), which initiates the extrinsic coagulation pathway. Monocytes may be able to downregulate PSGL-1 expression in response to platelet binding, but this has not been definitively established [143]. HIV transactivating factor (Tat) interacts with platelet integrin β3 and CCL3, resulting in the secretion of soluble CD40 ligand (sCD40L), a platelet activator [146]. Subsequently, activated platelets initiate the formation of PMA [144,145]. The CD40L released, also increases B cell activation and secretion of immunoglobulins in vitro and in vivo in mouse models, which has been associated with ITP in non-HIV infected individuals and could play a similar role in HIV positives [147]. Platelets isolated from HIV patients secrete significantly more sCD40L in response to stimulation compared to healthy controls. The same effect was seen for other proinflammatory molecules, such as CCL5 and P-selectin expression. Patients who are not on c-ART can also have diminished responses however, which is attributed to exhaustion [148]. Overall, this evidence suggests that HIV infection sensitizes platelets to procoagulant and proinflammatory stimuli, which in response release more procoagulant and proinflammatory factors and induce monocytes to do the same. This creates a positive feedback loop underlying chronic inflammation and risk of thromboembolic complication. Although initiation of c-ART normalizes both platelet and monocyte activation markers in plasma within 12 weeks [149], these studies using isolated platelets and monocytes demonstrate that virological suppression seems unable to completely disrupt this inflammation-coagulation cycle present in people living with HIV.

### 3.4. Rodent-Borne Viruses

#### Hantavirus

Hemorrhagic Fever and Renal Syndrome ***(HFRS)*** caused by Eurasian hantaviruses is associated with severity-dependent thrombocytopenia. The clinical characteristics of the disease include both hemorrhagic and thrombotic complications. This suggests that platelet consumption plays a significant role in the pathogenesis of thrombocytopenia. A longitudinal study of 35 hospitalized HFRS patients with Puumala virus (PUUV) infection revealed a biphasic pattern in platelet counts over time. This started with marked thrombocytopenia upon first presentation to the clinic, followed by a rise to just below the upper limit of normal approximately 2 weeks after symptom onset, and subsequent normalization of platelet counts. Elevated plasma TPO levels, reticulated platelet counts and MPV (each peaking at the platelet count nadir) suggest a compensatory myeloproliferative response to an acute loss of platelets within the circulation during early infection. Ex vivo expression of platelet surface activation markers was higher during the acute disease stage compared to follow-up, although the opposite was found for in vitro stimulation assays, indicating platelet exhaustion. Finally, patients who had signs of Disseminated Intravascular Coagulation (DIC) or thrombosis during the disease had higher plasma platelet activation markers, such as soluble P-selectin and soluble Glycoprotein IV (GP IV) compared to those who had not [150]. Another longitudinal study measured Von Willebrand Factor, Fibrinogen, fibronectin and A Disintegrin Additionally, Metalloproteinase with a ThromboSpondin type 1 domain (ADAMTS13) concentrations in plasma of PUUV infected patients during the acute and recovery phase. These factors where indeed elevated during acute disease when compared to recovery. Thrombocytopenia was present in 15 out of the 19 patients studied, but the exact numbers and how they correlate with the markers measured in plasma is not described [151]. Furthermore, during the acute phase of PUUV infection platelet aggregation appears impaired, especially when induced with thrombin, when tested on impedance aggregometry. Platelet adhesive mechanisms on collagen are intact, despite thrombocytopenia, while thrombopoiesis is active [152]. Potential mechanisms explaining the decrease in platelet count based on In vitro data include decreased production due to bone marrow invasion and megakaryocyte infection [153] and binding of platelets to infected endothelial cells [154].

### 3.5. Gastrointestinal Tract Viruses

#### 3.5.1. Enteroviruses 

Since the near-eradication of poliomyelitis, severe enterovirus infections have become rare, typically causing mild cold-like illness in children and hand-foot and mouth disease. Coxsackievirus infections occasionally cause myocarditis in adults, and Enterovirus 68 has recently been implicated in episodes of acute flaccid myelitis in children. Thrombocytopenia has been described in neonatal cases of coxsackievirus B3 (CoxV B3) infections, both in Japan, with one case being related to secondary Hemophagocytic Lymphohistiocytosis (HLH) triggered by CoxV B13 [155,156]. A cross-al study of neonatal cases with severe enteroviral infection in Japan demonstrated significantly decreased platelet and WBC counts in Human Parechovirus 3 infections compared to RSV infected controls. In addition, the Human Parachovirus 3 infected patients showed elevated plasma ferritin and lactate dehydrogenase (LDH) levels, both compared to the RSV controls and infants with other enterovirus infections. This may suggest an HLH-like illness secondary to this viral infection [157]. Human platelets express the Coxsackie-Adeno Receptor (CAR), aCoxVB3 cell entry receptor. Whether this mechanism plays a role remains unclear, in particular as the virus appears to be unable to replicate in platelets in vitro. However, platelets become activated after incubation with CoxVB3, increasing their expression of P-selectin and showing signs of apoptosis (i.e., increased phospatidylserine (PS) expression) [158].

#### 3.5.2. Rotavirus (RotV) 

Viral gastroenteritis is a major cause of child mortality in the developing world and a significant burden on the healthcare system in developed countries [159]. Outbreaks of norovirus and rotavirus frequently occur in childcare institutions and nursing homes, where those most vulnerable to dehydration are affected. These infections are typically limited to the gastrointestinal epithelium and rarely cause severe systemic disease with involvement of multiple organs or severe inflammation. It is therefore not surprising that thrombocytopenia is not a dominant clinical feature of viral gastroenteritis. Mean platelet counts in children with rotavirus gastroenteritis are reported within the normal range and not diverging from children with other viral causes of gastroenteritis [160]. It is important to recognize both concentration and dilution effects of intravascular fluid shifts when assessing platelet counts or other blood cell counts in patients with conditions such as severe gastroenteritis or other critical illnesses.

### 3.6. Herpesviruses

#### 3.6.1. Cytomegalovirus (CMV)

CMV is present in nearly all humans and remains latent for life after primary infection, which is usually asymptomatic in immunocompetent hosts. In contrast, severe immune suppression can lead to reactivation in later life leading most notably to CMV-mediated enterocolitis, hepatitis, retitinis or encephalitis [161]. However, very little is known about platelet counts during primary infection in immunocompetent hosts, because CMV infection comes to the attention of clinicians only when these rare complications develop. In vitro, CMV has been shown to interact with platelets through TLR-2. However, rather than directly resulting in platelet aggregation, these activated platelets produce a proinflammatory response, form aggregates with leukocytes and increase their adhesion to human vascular endothelial cells (HUVEC). Thus, in this model of CMV infection, platelets act as an intermediary between the virus and circulating immune cells [39].

#### 3.6.2. Epstein Barr Virus (EBV)

Like all other human herpesviruses, infection with EBV occurs in the majority of the population at an early age. This leads frequently to a mild but sometimes protracted viral symptomatic episode called infectious mononucleosis, which is followed by lifelong latency. Occasionally reactivation occurs in immunocompromised hosts, most commonly in organ transplant recipients. Recently, primary EBV infection has been associated with a variety of auto-immune diseases, whereas latent EBV infection and reactivation plays a role in the pathogenesis of Hodgkin’s lymphoma and B and T cell lymphoma’s, through poorly understood mechanisms. EBV infection is also a well described trigger for secondary Hemophagocytic Lymphohistiocytosis (HLH) [162].

In the case of EBV reactivation, the relation with lymphoproliferative disorders is important to keep in mind when evaluating a patient with thrombocytopenia, especially when other cell-lineages are involved.

In cases of primary EBV infection, thrombocytopenia and hemolytic anemia are occasionally also found and have been associated with the presence of platelet and erythrocyte auto-antibodies. Typical of primary EBV infection is the production of heterophile antibodies by naïve B cells that have become infected with latent-phase EBV [162]. Some of these antibodies may be autoreactive and bind to platelets, leading to their destruction. Due to EBV’s restriction to human hosts, well established animal models to study viral-platelet interactions in vivo do not exist. However, experimental infections with the related murine gammaherpesvirus 68 (γHV68) produce a mononucleosis-like illness in mice. This shows a significantly reduced platelet count during the early latent replication phase (nadir 17 days post infection). In this model, thrombocytopenia was found to be the result of antibodies induced by the infection and depended on viral latency, supporting the notion that polyclonal antibodies produced by latently infected B cells include autoantibodies against platelets [163]. This mechanism appears to be unique to EBV infection and is separate from auto-antibodies induced by other viral infection, which is believed to be the result of molecular mimicry between viral and self-antigens.

#### 3.6.3. Human Herpesvirus 6 (HHV-6) 

HHV-6 causes a near universal childhood illness, exanthema subitum, before entering its latent stage. Reactivation is rare, and generally only occurs during profound immunosuppression, such as during allogenic hematopoietic stem cell transplantation. In this population, HHV-6 reactivation (defined as a positive PCR on blood samples) was significantly associated with delayed platelet engraftment and the development of graft versus host disease (GVHD) [164].

#### 3.6.4. Varicella Zostervirus (VZV)

Primary VZV infection almost universally presents itself as a self-limiting childhood illness, with more significant sequalae emerging later in life, ranging from common herpes zoster to rare cases of severe disseminated disease. The latter is typically only found in immunocompromised hosts, although not exclusively [165]. Typical Herpes Zoster manifests itself as a vesicular cutaneous eruption restricted to one dermatome and is most frequently seen in the elderly and patients who received chemotherapy for solid or hematological malignancies. Herpes Zoster can have long-term sequalae, such as post-herpetic neuralgia, but does not cause systemic disease and patients usually have normal platelet counts [166]. Profound thrombocytopenia has been described in reports of disseminated VZV infection, combined with DIC, hemorrhaging, ischemic strokes, ileus, abdominal pain, hepatitis, meningoencephalitis and vasculitis [167,168,169,170]. The vasculitis is believed to be caused by VZV infection of the arterial walls themselves and can be found in arteries in various organs, including smaller cerebral arteries, where it is associated with stroke [171]. Case series describing VZV related strokes report elevated platelet activation markers, such as PF-4 and β-thromboglobulin levels in some patients [172]. Splenomegaly with associated hypersplenism is a common feature of systemic herpesvirus infections, which contributes to thrombocytopenia and sometimes leads to splenic rupture [173]. The differential diagnosis of thrombocytopenia during a VZV infection is broad, because of comorbidity-related immune suppression, and includes immune thrombocytopenia, drug induced thrombocytopenia [174] and bone marrow dysfunction, particularly if other lineages are affected. First presentation or relapse of ITP has been reported during primary VZV infection in adulthood [175,176]. A platelet count <200 × 10^9^/L was found to be predictive of a poor outcome in patients suffering from Ramsey Hunt Syndrome [2]. A large prospective cohort study identified thrombocytopenia as an independent risk factor for ICU admission in hospitalized children with VZV infection, although the rate of underlying hematological comorbidities and bacterial coinfections was high, suggesting VZV was not the sole cause of this phenomenon [177].

### 3.7. Respiratory Tract Infections

A platelet count close to the lower reference limit is a common finding in more severe viral respiratory tract infections [178]. Data on platelet counts in mild respiratory infections are scarce, possibly due to the fact that these cases typically do not present to care and blood counts are rarely performed. Interestingly, the literature reports thrombocytosis in infants hospitalized with respiratory tract infections, especially RSV and rhinoviruses, with platelet counts decreasing with age [179,180,181]. Platelet counts in patients with acute exacerbations of heart failure who tested positive for respiratory viruses by PCR, did not differ significantly from those who tested negative [182]. The lungs have recently been found to host resident megakaryocytes, which contribute to platelet production [183]. Investigating the interactions between respiratory viruses and platelets could be key to understanding the high rate of thromboembolic complications that arise during viral acute respiratory distress syndrome (ARDS).

#### 3.7.1. Adenoviruses (Adv) 

Among viruses causing mild upper respiratory tract infection, adenoviruses appear to be most studied in relation to platelets. Coxsackie and Adenovirus receptor (CAR) is the receptor adenoviruses use for cellular attachment. Expression of this receptor has been reported in healthy human platelets, albeit at a very low frequency (3.5%) [184]. In vitro studies where platelet rich plasma was incubated with very high concentrations of human adenovirus 3 and 5, showed a moderate increase in platelet aggregation and platelet activation marker expression, with uptake of adenovirus 5 by platelets demonstrated using EM [185,186]. Since natural infection in humans is unlikely to expose platelets to the high viral titers used in these incubation experiments, the clinical relevance for this finding is mostly related to the potential future use of adenoviruses for gene therapy purposes. Indeed, cancer patients experimentally treated intravenously with oncolytic adenovirus where serially sampled to determine relative abundance of viral DNA in various blood cell populations. Although very little platelet-associated virus was found in vivo, in vitro experiments where whole blood was incubated with the studied adenoviruses revealed a large proportion of virus bound to platelets. Given the thrombocytopenia observed during adenovirus-based treatments, this discrepancy could be the result of a survival disadvantage of adenovirus bound platelets in the circulation [187].

#### 3.7.2. Influenza Virus (IAV/IBV) 

Influenza virus infection is associated with a severity-dependent thrombocytopenia. Pediatric outpatients with confirmed IAV or IBV infection showed slightly, though significantly, lower mean platelet counts compared to asymptomatic controls. Children with influenza-like illness who were IAV and IBV PCR negative had platelet counts in between the confirmed positive and healthy groups, and platelet counts could not reliably distinguish between influenza positive and negative children [188]. In adults, severe influenza infection is accompanied by an increased risk of pulmonary thromboembolisms and cardiovascular events (Sellers, 2017 #1) suggesting platelet activation occurs during infection. Whole-blood transcriptome studies have found gene expression signatures in patients during H1N1 infection that are associated with a poor response to antiplatelet agents. Conversely, patients undergoing coronary catheterization that had a gene expression signature associated with viral infection, where more likely to have a confirmed myocardial infarction compared to those that did not express this signature [189]. Pathogenic H3N2 and H1N1 strains are capable of infecting pulmonary vascular endothelial cells, which increases platelet adhesion to both infected and nearby uninfected cells through interaction between endothelial fibronectin and platelet integrins [190]. Various influenza A strains cause thrombocytopenia in experimentally infected ferrets, with highly pathogenic strains (H5N1) showing a stronger decrease compared to moderate (H1N1) or mildly pathogenic (H3N2) strains. In addition, these viruses are capable of directly infecting platelets in vivo through binding of sialic acids on glycans on their cell surface. EM imaging has demonstrated the ability of platelets to phagocytose influenza virus particles. This infection of platelets results in their activation, aggregation and subsequent clearance from the circulation. Interestingly, desialylation of platelet glycans by viral neuraminidase is hypothesized to reduce the lifespan of affected platelets through increased hepatic clearance [191]. Influenza virus can also interact with platelets through TLR7, which leads to the formation of platelet-neutrophil aggregates and neutrophil NETosis, through complement (C3) secreted by the platelets [192]. Immune-complexes of antibodies against influenza virus are also capable of activating platelets through an interaction with the Fc-ϒIIA receptor present on the platelet surface, leading to thrombocytopenia in a humanized mouse model. These findings, combined with reported influenza vaccine induced ITP, point to a link between influenza virus-specific adaptive immunity and thrombocytopenia [193,194,195].

#### 3.7.3. Measles Virus (MV) 

Likely the most contagious virus known to affect humans, this virus first infects the respiratory tract and subsequently spreads to lymphoid organs, infecting lymphocytes, including memory B and T cells [196]. A highly effective vaccine has been available for several decades, yet immunization programs have not been able to reach sufficient coverage to eradicate the disease, leading to sporadic outbreaks [197]. Studies published in the past 10 years describing natural infection in adults report mild leukocytopenia and thrombocytopenia as a frequent finding, occasionally with minor bleeding complications, but no thromboembolisms [198,199,200]. A link between Subacute Sclerosing Panencephalitis (SSPE), a late complication of Measles caused by persistence of MV in the brain, and ITP has been proposed, based on the co-occurrence of both extremely rare diseases in 3 pediatric cases [201]. This is further supported by an increased incidence of ITP after MV vaccination, where platelet binding anti-MV (and anti-rubella) IgG and IgM was demonstrated [202].

#### 3.7.4. Parvovirus B19 (PVB-19)

While best known as a mild, self-limiting childhood illness (fifth disease), PVB-19 can occasionally cause more severe disease, especially during pregnancy, resulting in hydrops fetalis. Due to its tropism for erythroid progenitor cells and megakaryocytes, fetal PVB-19 infection causes severe anemia and thrombocytopenia, requiring Intrauterine Transfusion (IUT) of platelets and erythrocytes in some cases [203,204]. While most severe and best described in fetal infections, PVB-19 can also cause thrombocytopenia, anemia, leukopenia or pancytopenia in children and adults [205,206,207,208]. A retrospective cohort study reports PVB19 infection in children undergoing chemo- and radiation therapy for non-hematological malignancies increases the risk of thrombocytopenia and transfusion of blood products [209]. A similar study comparing malignant and nonmalignant hematological disease in a pediatric population found that PVB-19 DNA positivity was not associated with a higher risk of transfusion, but the number of platelet transfusion units administered per patient was over 3-fold higher in PVB-19 DNA positive patients [210]. PVB-19 is also able to infect myocardial tissue, leading to clinical myocarditis and dilated cardiomyopathy. This raises the question whether PVB-19 is also capable of infecting vascular endothelium and cause vasculitis and platelet adhesion to infected vessel walls. A case–control study exploring differences in microparticle (MP) profiles in the peripheral circulation of patients with myocarditis caused by PVB-19 versus other causes, found significant increases in apoptotic endothelial, platelet and leukocyte-derived MPs in PVB-19 mediated disease. This suggests that, in addition to impaired hematopoiesis, PVB also causes platelet destruction and vascular damage [211]. In vitro studies suggest PVB-19 nonstructural protein 1 (NS1) causes endothelial activation, upregulation of adhesion molecules and an increase in platelet and monocyte binding [212].

#### 3.7.5. Respiratory Syncytial Virus 

Severe respiratory infections with RSV occur mainly in children, the immunocompromised and those with underlying pulmonary disease [213]. In contrast to other respiratory infections, thrombocytosis rather than thrombocytopenia appears to be a common phenomenon found during acute RSV disease [181]. In vitro experiments demonstrate a reduction of monocyte RSV infection when platelets are added to the culture, possibly by binding and internalization of RSV. Platelets increase surface P selectin expression in the process, but why this would lead to thrombocytosis rather than thrombocytopenia is unclear [214].

#### 3.7.6. SARS Coronavirus 2 (SARS-CoV-2, COVID-19)

The literature cited in this part of the review was updated shortly before submission to include the high volume of scientific work that has been published on this virus, which has caused a pandemic of severe pneumonia of historical proportions. Besides bilateral pneumonia, critical COVID-19 cases are characterized by multi-organ disease [215], and a remarkably high incidence of pulmonary embolisms [216,217]. Several mechanisms involving hypercoagulability and inflammation interact resulting in thrombotic phenomena both in the microvasculature and in the larger, mostly pulmonary blood vessels [218].

In fact, upon autopsy these embolisms were found to be mainly composed of platelets, fibrinogen and neutrophils [219,220,221]. Another typical finding during autopsy of deceased COVID-19 patients is the presence of widespread microvascular thrombosis in both pulmonary and extrapulmonary vessels, including in patients without true thromboembolisms, indicating a systemic prothrombotic state [220].

A low to low-normal platelet count is present during peak symptomatic illness, with increased MPV and PDW, and expression of surface activation markers [222,223]. However, one study identified subpopulations of platelets with a downregulated phenotype, which were highly enriched in severe, but not in intermediate COVID-19 cases, suggesting exhaustion of circulating platelets [220]. The total platelet population from these patients still showed hyperresponsiveness to procoagulant stimuli in vitro, likely driven by a hyperactive minority that was also present. This hyperresponsiveness was also found in other studies [224]. Some clinical studies report that thrombocytopenia is associated with increased mortality [4,5], whereas other do not [215,225]. This discrepancy might depend on disease severity, comorbidities or the type of care provided. For example, a well-defined cohort of mechanically ventilated critically ill patients showed that daily platelet concentrations were not associated with intensive care unit survival [215]. However, this observation does not exclude a role for platelet (dys)function in immunothrombosis. Other coagulation-related markers during acute illness show strongly elevated levels of D-dimers and fibrinogen degradation product (FDP), normal to slightly prolonged PT, APTT, elevated plasma viscosity and coagulability and normal to mildly increased INR [5,12,222,223]. Whether these markers have diagnostic or prognostic value requires investigation and might differ along the course of infection depending on disease severity, comorbidities and type of care provided. Platelet counts appear to rise slowly over the course of the disease, which coincides with a sharp peak in IL-6, suggesting this cytokine may play a role in the thrombopoietic response [220]. Plasma TPO levels are elevated in severe COVID-19 patients, but gene expression of its receptor, c-MPL is decreased, suggesting desensitization of the bone marrow as an additional mechanism for thrombocytopenia in COVID-19. When thrombocytopenia is present, it is often accompanied by relative deficiencies in other myeloid and lymphoid cell lineages [226,227], which could indicate bone marrow displacement caused by a proliferative response to hyperinflammation, either as a toxic effect of cytokines to progenitor cells in the bone marrow or a result of homing to inflamed tissues and extravasation.

Considerable work within a relatively short timespan has been done unraveling the mechanisms through which SARS-CoV-2 infection causes platelet activation. One study shows that platelet activation in severe COVID-19 is associated with detectable viral RNA in blood. Furthermore, the viral Spike protein enhanced platelet activation, aggregation, thrombus formation and degranulation in vitro and in a mouse model. This effect was only seen when the full Spike protein or its ACE-2 binding S1 subdomain were used, not the S2 domain. This suggests that ACE-2 signaling mediates this platelet activation. Further analysis of intracellular messaging points towards involvement of the MAPK signaling pathway. The same study confirmed expression of ACE-2 in human platelets using immunofluorescence [223]. However, another study did not detect any ACE-2 mRNA or protein expression in COVID-19 patients by RNA-seq, qPCR or Western blot [224]. To date, no study has demonstrated SARS-CoV-2 internalization by platelets.

It is also clear that platelets influence the host immune response to SARS-CoV-2. Platelet gene expression profiles in severe and critically ill COVID-19 patients showed shared pathways with sepsis and Influenza H1N1 infection. These show related antigen presentation and immune regulation, including differential expression of interferon-induced transmembrane protein 3, which has antiviral properties [224]. The formation of platelet-leukocyte aggregates was also found, with neutrophil-platelet aggregates correlating with the severity of lung injury and leading to the formation of NETs [220,221,224,228,229]. Similar to observations in HIV and DENV, platelet-monocyte complexes are formed in severe COVID-19 patients via platelet P-selectin, which results in overexpression of Tissue Factor on the monocyte surface, the key initiator of the extrinsic coagulation pathway [230].

As with many other viral infections, reports have been published of cases of ITP associated with COVID-19 infection, including one case of Evans syndrome [231]. Another case report illustrates the importance of performing a peripheral blood smear in COVID-19 patients with severe thrombocytopenia to exclude EDTA dependent pseudo-thrombocytopenia [232].

## 4. Conclusions

The topic of thrombocytopenia in viral infectious diseases has been actively studied for many decades, with the last 10 years yielding many new insights. A scientific field combining the disciplines of virology, hematology and increasingly immunology is revealing a complex system of interactions between various viruses, the coagulation cascade and the innate and adaptive immune system. Increasingly, platelets are regarded as part of the immune system, in addition to being capable of forming blood clots. The rapidly changing world of viruses ensures that this field is constantly forced to adapt to new outbreaks and is therefore equally dynamic. The current COVID-19 pandemic has brought platelet-virus interactions to the forefront, with many publications addressing this topic being available within a year after the SARS-CoV-2 virus first emerged.

The absence of research on the “classical” hemorrhagic fevers, such as Ebola, Lassa and Marburgvirus, has been notable however, despite two large outbreaks of Ebola occurring in the last decade. The high level of biological containment required to safely study these viruses, combined with the extremely resource-limited settings in which these outbreaks occurred make doing research into these viral diseases extremely challenging. Nonetheless, significant progress has been made in preventive and therapeutic interventions for Ebolavirus, with the successful trials of several vaccines, [233] antiviral drugs and monoclonal antibodies [234].

DENV, another viral infection disproportionately affecting people in resource limited settings, was the virus we found most publications about in relation to platelets in the past decade. This is not surprising, given the considerable role platelets play in the pathophysiology of severe disease and the enormous public health burden associated with the virus. Despite the considerable knowledge gained, this has so far not been translated into clinically effective interventions. However, we did find several studies investigating therapeutics aimed at modifying platelet function in DENV infection, which will hopefully bear fruit in the coming decade.

Looking in detail at the interactions between viral infections and platelets revealed several common pathways connecting inflammation and platelet activation, which has been termed “Immunothrombosis”. This is a term which has not yet been clearly defined as a clinical or pathological entity and has some overlapping features with DIC, with the main clinical difference being the absence of significant bleeding. COVID-19 may provide us the opportunity to increase our overall understanding of thrombocytopenia in viral infections and perhaps to study a new dimension of immunothrombosis which could be translated to other viral infections. It is especially important to gain more understanding about which interventions could aid in reducing the morbidity and mortality related to immunothrombosis. As platelet dysfunction is often accompanied by an increased risk of both bleeding and thrombosis, approaching this issue with conventional anticoagulants often involves having to choose the lesser of two evils. Immunomodulation therapy is a rapidly evolving field, with many newly available therapeutics, most of which have not yet been trialed in viral infectious diseases. This approach warrants further study, but here caution is also advisable, given the possibility that some mechanisms involved in immunothrombosis are required in the antiviral response in the host.

## Figures and Tables

**Figure 1 jcm-10-00877-f001:**
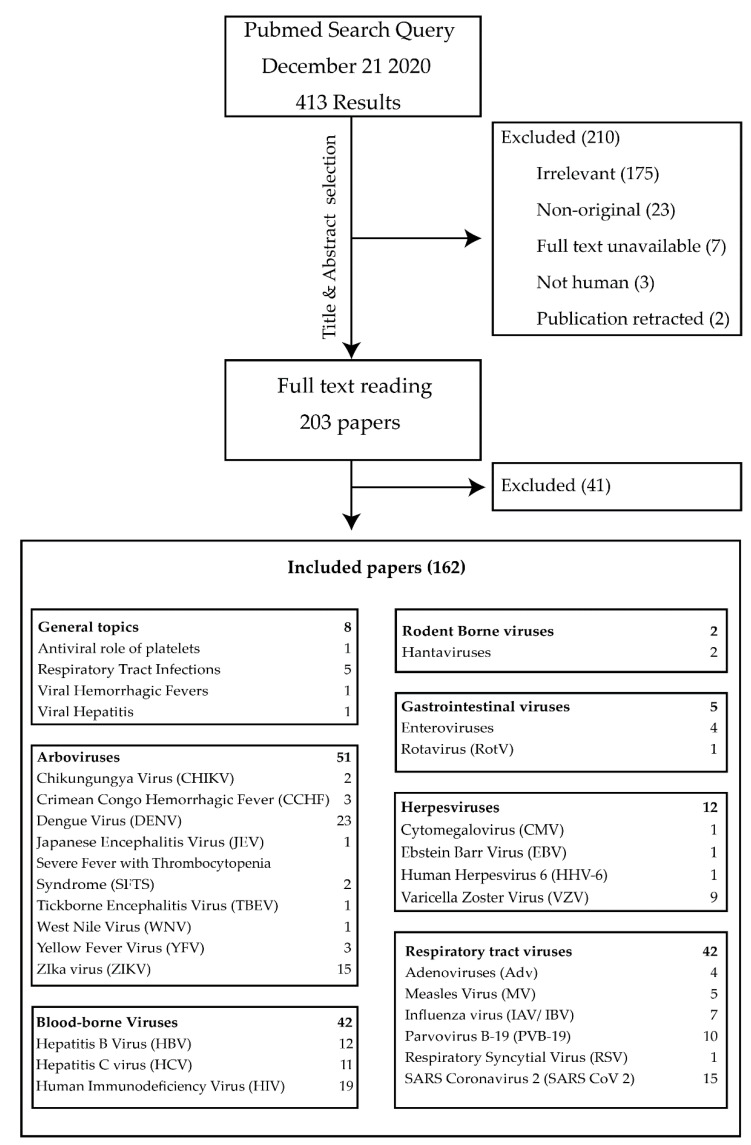
Flow diagram of PubMed search results.

**Table 1 jcm-10-00877-t001:** Overview of mechanisms contributing to thrombocytopenia in a selection of major viral infections. U = unknown.

Virus	Platelet Binding (Receptors)	Platelet Activation (Receptors, Markers)	Platelet Infection (Receptors)	Platelet Replication	Vascular Endothelial Disruption	Impairs Platelet Production	Associated with Hemorrhage	Associated with Thrombo-embolism	Platelet Sequestration	PLA Formation	Autoimmunity
DENV	DC-SIGN, HSP, FcϒR2A, GPIb, CLEC2, CLEC5A [66,67,81]	Receptors: CLEC2, CLEC5A, TLR 2/4, 5HT2 [66,67,71] Markers: P-Selectin, Integrin α2b, PF4, CCL5, PS, CD40L, CD63, GP1b, GPIIb/IIIa, microparticle release, aggregation [63,64,65,66,67,70,73]	DC-SIGN, HSP, FcϒR2A (Ig bound virus) [30]	Yes [30]	Plasma leakage [59,67]	Infects megakaryocyte like cells [80,81]	With endothelial and platelet dysfunction [56,57,58]	Rare [235]	Phagocytosis [73,76]	PMA, PNA, inducing NETosis [68,69]	Immune complexes [75]
ZIKV	U	U	No [116]	No [116]	Infects vascular endothelial cells in vitro [236]	U	In severe cases [107,108]	Rare [237]	U	U	ITP [112,113,114]
YFV	U	U	U	U	Fundoscopic abnormalities [101]	U	Hepatic failure and deficiency of plasma coagulation factors. [99,100] Cirrhosis associated varices (HBV/HCV)	Microvascular thrombosis [101]	U	U	U
HBV	U	Markers: Morphological changes [127] resistance to antiplatelet agents [128]	U	U	Polyarteritis nodosa (rare) [238]	Impaired hepatic TPO production [131,239]		Increased risk of VTE [240] Portal vein thrombosis in cirrhosis [241]. Risk of ischemic cardiovascular disease elevated for HCV only [242,243]	Portal hypertension and hypersplenism [239]	U	U
HCV	Likely, Mechanisms unknown [130,132,133]	U	Greater stability of HCV in platelets, [130,132] persistence in platelets during treatment [133]	No [132]	Endothelial activation [244], capillarization of liver sinusoidal endothelial cells [245], (non)cryoglobulinemic vasculitis [246]					U	ITP [130]
HIV	DC-SIGN, [138] GPIIIa, CCL-3 [146]	Receptors: CXCL4, CCR3, GPIIIa. Markers: PF-4, CCL5, P-selectin, sCD40L, CCL5, Conflicting reports on morphological changes, oxidative stress.(135–137, 139)	Yes, via megakaryocyte precursors [141]	No	HIV-associated vasculopathy [247], Infects arterial smooth muscle cells [248]	Infection and impairment of hematopoietic progenitor cells [249]	No	Increased risk of VTE [250], myocardial infarction [251] and cerebrovascular disease [247]	Hyper-splenism [252]	PMA, increasing monocyte TF expression [144,145]	ITP [253], HLH [254]
IAV/IBV	Likely NA binding of glycans [191]	Receptors: Glycans, TLR7, [44] Immune complexes via Fc-ϒIIA receptor [193] Markers: P-selectin, [191] CD40L, C3, CD63[192] Resistance to antiplatelet agents.[189]	Phago-cytosis [191]	U	Infection of pulmonary vascular endothelial cells [190]	U	No	VTE, myocardial infarction, ischemic cerebrovascular accidents [1]	U	PNA, inducing NETosis [192]	ITP [194,195]
SARS CoV 2	ACE2, conflicting evidence on platelet expression [223,224]	Receptors: ACE2, TLR4, CLEC2, CXCR-4. Markers: P-selectin, GPIb, GP IIa GPIIIb, GPIIb/IIIa, CD40L, CD63, morphological changes, aggregation, degranulation [220,222,223,224]	U	U	U	Decreased cGMP expression [224]	No	Microvascular thrombosis and VTE [216,217,218,219]	U	PMA increasing monocyte TF expression, PNA inducing NETosis [221,230]	ITP, Evans Syndrome [231]

## Data Availability

Data available in a publicly accessible repository. The data presented in this study are available via the national Center of biotechnology information.

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
