# Peer review of "Thrombocytopenia in Virus Infections"

_jcm, 2021, doi:10.3390/jcm10040877_

Round 1

Reviewer 1 Report

The authors present a review about the potential causes for thrombocytopenia during viral infections and the putative role of platelets in the immune response to viral infections.

The review is appropriate, well organized, well written and it exposes the data concisely and clearly. It exhaustively compiles the literature available about the topic, with a procedure for literature searching and selection that seems appropriate. All references are appropriate and relevant. The topic under review is important for the field not only from the perspective of basic research but also from the perspective of infection control and treatment of human infections.

I only have minor comments/suggestions:

  1. There are a couple of typos that should be checked: 
  • Line 10: in "150 x 109/L", "L" is duplicated here.
  • Line 72: "in" instead of "or" should be used here.
  • Line 665: "PVB13" is mentioned here instead of "PVB19".
  1. Some additional references might be added at some points in the text, mainly for background context and to prompt the reader in the way for expanding about certain subtopics that are not the focus of the review but are mentioned in it. Below I am listing where in the text (by line) these references might be added:
  • Introduction being short, condenses much information clearly and concisely. In general, it would benefit from some more references. I suggest: Lines 39, 42-43, 54, 57, 59 and 64. 
  • Line 117 (reference 16): this paper covers the role of platelets in the immune response to bacterial infections but do not mention viruses at all. I suggest looking for a more appropriate reference. 
  • Lines 242, 249, 501, 514, 538, 595, 649, 684, 703, and 776.  
  • Line 308 (reference 73): is there in the literature a study that covers more that only 2 cases?
  • Paragraph between lines 523 and 531 needs appropriate references.
  • Paragraph between lines 730 and 739 needs appropriate references.
  1. A glossary of abbreviations might be added to the manuscript. Most of them are defined in the text after first use. However, the review is long, and because of the nature of the topic, it contains many of these. I believe such a glossary will help the reader to better follow and understand the manuscript, especially for those with a non-medical background. 

Author Response

Dear Reviewer,

Thank you for your valuable comments. 

Here are our responses:

  1. The typo's you have mentioned have been corrected.
  2. References: 
    1. Line 39: The claim that the incidence of thromboembolism is elevated after viral infection, has been made more specific for influenza virus, as it is best documented here. An appropriate reference (literature review) has been added on this topic. 
    2. Line 42-43: The sentence has been adjusted to reflect that platelet counts are more a marker of disease severity (which indirectly affects prognosis) rather than a predictor of outcomes. Several references have been added, which where already discussed in more detail in the main text. 
    3. Line 54: A reference to a review on the role of cytokines in megakaryopoiesis was added. 
    4. Line 57-59: The fact that cytokines have a role in inflammation proliferation of leukocytes we consider to be common knowledge.
    5. Line 64-66: 2 references added describing mRNA in platelets (Review) and dengue virus replication in platelets (already included in original submission). 
    6. Line 117: a more appropriate reference was found, which specifically discusses viral, rather than bacterial infections. 
    7. Line 242: A review on the clinical aspects of Japanese encephalitis was added to provide more context. 
    8. Line 249: Review article on SFTS was referenced for more context. 
    9. Line 501: A review on the global burden of rota virus infection was added. 
    10. Line 514: CMV reference added
    11. Line 538: EBV reference added
    12. Line 595: Reference to megakaryopoiesis in the lungs added. 
    13. Line 649: Measles virus reference added
    14. Line 684: RSV reference added
    15. Line 703: Added appropriate references to statement about elevated platelet activation markers in COVID-19. 
    16. Line 776: References do Ebola virus vaccines and therapeutics added. 
    17. Line 308: Added another reference which describes coagulation abnormalities in YF in larger cohort. 
  3. A glossary of abbrevations was added. 

Reviewer 2 Report

Raadsen et al. present a complete and very well-organized revision of current literature on the effects of viral infection on platelet count, function and production. The paper is well written and structured, and it represent the first comprehensive literature review on the topic addressed by the authors. It touches all the most relevant topic in the field. It is an exhaustive and helpful. In this reviewer opinion, it could benefit from a paragraph on potential therapeutic approaches targeting platelets during these viral infections.

Author Response

Dear Reviewer,

Thank you for your valuable comments. 

Unfortunately, we found too few studies on therapeutic approaches for thrombocytopenia or platelet dysfunction in viral infections to base a paragraph on. This is mentioned in the conclusion section of the review. In the final paragraph, we discuss the need for future research into immune- and coagulation modulating drugs to prevent and treat bleeding and thromboembolic complications during or after viral infections.

Reviewer 3 Report

This is an extremely thorough and comprehensive review on thrombocytopenia and viral infections. While, unfortunately, many of the proposed mechanisms are speculative, this reflects the current state of understanding in the field. 

The paper is thorough, but long, dense and a bit hard to read. It would benefit greatly from a few figures demonstrating the more common proposed mechanisms for specific virus graphically. Or perhaps showing different common mechanisms with notation as to which viruses use which pathways. 

Author Response

Dear Reviewer,

Thank you for your insightful comments on the paper.

We agree that some figures could improve the readability of this paper. 

We chose not to include them however, for the following reasons:

  1. Graphical representations of the mechanisms of platelet-virus interactions can be found in many other papers on this topic and therefore including them in this paper would not be original.
  2. We have intentionally chosen for an approach which is as thorough and systematic as possible on this subject which, to our knowledge, has not been done recently. The amount of information contained in this review is therefore substantial and difficult to capture in figures. As the reviewer mentioned, many of the mechanisms proposed are still speculative and figures risk obscuring this fact. 
  3. There is currently insufficient time left before the publication deadline of this paper to create new figures that is both original and adequately capures all information contained in this review. 

Reviewer 4 Report

The manuscript is an excellent review of the relationshipbetween thrombocytopenia and virus infection in humans. It describes the mechanisms of interaction between platelets and viruses which lead to decrease of platelet counts.

The manuscript is prepared well, but in my opinion, a schema or a table summarizingthe causes of thrombocytopeniain virus infectionscould be a good supplement.Also,the Authors mayconsider removingfrom the main text the viruses which do not induce thrombocytopeniaas adominant clinical feature,such as rotaviruses, cytomegaloviruses, HHV-6, respiratory syncytial virus.It may be a good idea to get this information together in a separate paragraph.

Minor comments:

There is no reference to Figure 1 in the text.

pp.7, line249-250

is DC-SIGN, which is expressed by human platelets, wasidentified as a receptor for SFTS”, should be rather DC-SIGN, which is expressed by human platelets, was identified as a receptor for Bunyavirus “.

Author Response

Dear reviewer,

Thank you for your comments.

A table as described by the reviewer has been partially finished abd was originally intended to be included with this review. The size of the table became very large however, due to the number of viruses and mechanisms that are discussed. If the timeframe for publication of this review allows this, we could consider finishing it and including it as a supplement. We expect we would be able to finish it in about 1 week. 

We agree that some of the viral infections described in this review do not have significant associations to thrombocytopenia or platelet dysfunction. We did include these viruses however, in order to preserve the systematic nature of the review and not have to exclude papers that we still believed contained relevant information about the topic. 

A reference to figure 1 has bed added to the first paragraph of the results section.  

Line 249-250 We have adopted the reviewers' suggestion to change SFTS to Bunya virus.